https://doi.org/10.1038/s41467-022-28586-0　　**OPEN**

# Molecular recognition of formylpeptides and diverse agonists by the formylpeptide receptors FPR1 and FPR2

Youwen Zhuang[1,4], Lei Wang [2,4], Jia Guo[1,3], Dapeng Sun [2], Yue Wang[1,3], Weiyi Liu[1,3], H. Eric Xu [1,3✉] & Cheng Zhang [2✉]

The formylpeptide receptors (FPRs) mediate pattern recognition of formylated peptides derived from invading pathogens or mitochondria from dead host cells. They can also sense other structurally distinct native peptides and even lipid mediators to either promote or resolve inflammation. Pharmacological targeting of FPRs represents a novel therapeutic approach in treating inflammatory diseases. However, the molecular mechanisms underlying FPR ligand recognition are elusive. We report cryo-EM structures of $G_i$-coupled FPR1 and FPR2 bound to a formylpeptide and $G_i$-coupled FPR2 bound to two synthetic peptide and small-molecule agonists. Together with mutagenesis data, our structures reveal the molecular mechanism of formylpeptide recognition by FPRs and structural variations of FPR1 and FPR2 leading to their different ligand preferences. Structural analysis also suggests that diverse FPR agonists sample a conserved activation chamber at the bottom of ligand-binding pockets to activate FPRs. Our results provide a basis for rational drug design on FPRs.

[1] The CAS Key Laboratory of Receptor Research, Shanghai Institute of Materia Medica, Chinese Academy of Sciences, 201203 Shanghai, China. [2] Department of Pharmacology and Chemical Biology, University of Pittsburgh School of Medicine, University of Pittsburgh, Pittsburgh, PA 15261, USA. [3] University of Chinese Academy of Sciences, 100049 Beijing, China. [4] These authors contributed equally: Youwen Zhuang, Lei Wang. ✉email: Eric.Xu@simm.ac.cn; chengzh@pitt.edu

The human formylpeptide receptor (FPR) family comprises of three members, FPR1, FPR2 (also named FPRL1), and FPR3[1]. They were initially discovered and characterized as pattern recognition G protein-coupled receptors (GPCRs) to recognize peptides with N-terminal formyl methionine, which are derived from invading pathogens or host mitochondria as pathogen- or damage-associated molecular patterns (PAMPs and DAMPs)[1–5]. FPR1 and FPR2 share a high sequence similarity[6], yet they display different preferences of formylpeptides. FPR1 is a high-affinity receptor for many short formylpeptides such as the prototypical formylpeptide fMLF (N-formyl-Met-Leu-Phe), while FPR2 prefers longer peptides or peptides with specific sequences such as the phenol-soluble modulin (PSM) family of formylated peptide toxins produced by *Staphylococcus aureus*[2,3,7]. Formylpeptides act on FPR1 and FPR2 to induce multiple pro-inflammatory events including chemotaxis of neutrophils and macrophages and generation of inflammatory cytokines and oxygen species[2]. FPRs and other closely related chemotactic GPCRs such as chemokine receptors and the complement C5a peptide (C5aR) belong to the γ-subgroup of rhodopsin-like Class A GPCRs[8]. With few exceptions, all of them primarily signal through G proteins of the heterotrimeric $G_{i/o}$ family.

In the past two decades, numerous studies have found that FPR1 and FPR2, especially FPR2, exhibit unusual functional promiscuity. They can recognize a variety of chemically distinct endogenous ligands including proteins and lipids besides formylpeptides and play multifaceted roles in inflammation[2,3,9,10]. Accordingly, recent studies have uncovered important roles of FPRs in host defense, inflammation-related cardiovascular and neuronal diseases, and cancers that are not related to formylpeptide recognition[11–15]. For example, FPR1 has been proven to recognize a non-formylated bacterial protein to function as the host cell surface receptor for the causative pathogen of plague, *Yersinia pestis*[16]. FPR2 can also be activated by a host-derived non-formylated protein, serum amyloid A (SAA), to induce acute type-2 inflammation[17]. In addition, FPR2 can sense multiple peptides and lipid ligands to selectively activate anti-inflammatory or pro-resolving pathways to induce the resolution of inflammation and tissue protection (15). Those FPR2 ligands include annexin A1 (ANXA1), a lipid-binding protein involved in the anti-inflammatory action of glucocorticoids, and a group of bioactive lipids known as the specialized pro-resolving lipid mediators (SPMs)[18–23]. In addition, ANXA1 can also signal through FPR1. The ANXA1-FPR1 signaling axis is important for the chemotherapy-induced anti-tumor immune responses[24]. It has been proposed that FPR2 can be targeted by ligands generated at different stages of inflammation to switch on or resolve inflammation in order to maintain a balanced inflammatory response (17). Biased FPR2 agonists that can actively resolve inflammation represent a novel therapeutic frontier for various inflammatory pathologies including asthma and cardiovascular diseases[15,21,25,26]. To this end, several synthetic FPR2 agonists including peptides and small molecules have been developed and evaluated in pre-clinical and clinical settings[20,27–29].

Previous studies suggested that functionally distinct ligands act on FPR2 at different regions to induce ligand-specific conformational changes[30]. We have reported a cryo-electron microscopy (cryo-EM) structure of the FPR2-$G_i$ signaling complex with a synthetic peptide agonist, WKYMVm, which revealed a heart-shaped ligand-binding pocket of FPR2. A crystal structure of engineered FPR2 with WKYMVm has also been reported[31]. To investigate the molecular mechanism underlying ligand recognition and activation of FPRs, we determined cryo-EM structures of the human FPR2-$G_i$ complex with three agonists, fMLFII as a formylpeptide[32], CGEN-855A as a synthetic anti-inflammatory peptide[33], and Compound 43 (C43 hereafter) as a synthetic non-peptide FPR agonist[34,35], as well as a cryo-EM structure of the

human FPR1-$G_i$ complex with fMLFII (Supplementary Fig. 1a). These structures together with mutagenesis studies provide unprecedented molecular insights into how formylpeptides and non-formylpeptide agonists act on and activate FPRs.

## Results

**Cryo-EM structure determination and overall structures.** We assembled nucleotide-free complexes of $G_i$-coupled FPR1 and FPR2 with different agonists (Supplementary Fig. 1b-d). To further stabilize the complexes, we also added an antibody fragment, scFv16, binding to the interface between $G_{\alpha i}$ and $G_\beta$[36] (Supplementary Fig. 1b-d). The structures of FPR2-$G_i$ complexes with fMLFII, CGEN-855A, and C43 were determined to global resolutions of 3.1, 2.9, and 3.0 Å, respectively, and the structure of the fMLFII-bound FPR1-$G_i$ complex was determined to 3.2 Å (Fig. 1, Supplementary Fig. 2 and 3, Supplementary Table 1). fMLFII is a potent formylpeptide agonist for both FPR1 and FPR2[32]. CGEN-855A is a non-formylated peptide agonist of FPR2 with 21 amino acids, which showed high selectivity for FPR2 over FPR1[33]. It has shown promising anti-inflammatory and tissue-protective effects in an animal model of myocardial infarction[37]. C43 is a potent small-molecule agonist of both FPR1 and FPR2[35]. It was initially suggested to be anti-inflammatory[34], but studies showing contradictory results were reported later[35]. Nevertheless, C43 is chemically similar to two newly developed non-peptide FPR2 agonists, Compound 17b[25] and BMS-986235[38] (Supplementary Fig. 1a), which showed pro-resolving and cardiac protective effects in multiple studies. To assemble the FPR1 and FPR2 complexes, we used a dominant-negative version of human $G_{\alpha i1}$ with mutations to decrease nucleotide-binding[39], rat $G_{\beta 1}$ and bovine $G_{\gamma 2}$, similar to that in our previously reported structure of the WKYMVm-FPR2-$G_i$ complex[40].

We modeled FPR2 residues from G21 to L317 in all three structures and FPR1 residues from G21 to L316 based on the cryo-EM density. The density maps of fMLFII in both FPR1 and FPR2 structures are clear enough to allow modeling of all five residues (Fig. 1a, d). For CGE-855A, only the C-terminal 8 amino acids from Gln14 to Met21 (residues in the peptide ligands are referred to by three-letter names, and residues in FPR1 and FPR2 are referred to by one-letter names hereafter) were modeled in the structure to fit the density (Fig. 1b). The relatively high-resolution map of FPR2-C43 allows us to unambiguously define the binding pose of C43 (Fig. 1c). In addition, we modeled palmitic acid molecules to fit the clear density observed near the intracellular loop 2 (ICL2) in both structures of fMLFII-bound FPR1 and FPR2 (Supplementary Fig. 4a, b).

**Conserved $G_i$-coupled modes for FPR1 and FPR2.** The overall structures of the FPR1 and FPR2 signaling complexes are very similar to each other, suggesting a highly conserved mechanism of $G_i$-coupling. Indeed, the intracellular regions of FPR1 and FPR2 can be well-aligned, and no significant differences are observed in the receptor and G protein interaction profile for FPR1 and FPR2 (Fig. 2a). This was not unexpected since residues at the core region of the 7 transmembrane helices (7-TMs) and the cytoplasmic region including intracellular loops 1–3 (ICL1-3) are highly conserved in these two receptors[40].

Structural comparison with the $G_i$-coupled neurotensin receptor 1 (NTSR1) at two conformational states[41] suggested that our structures of the FPR1- and FPR2-$G_i$ complexes represent the canonical state (Supplementary Fig. 4c). As observed in other GPCR and G protein complexes, the C-terminal half of α5 of $G_{\alpha i}$ inserts into the cytoplasmic cavities of FPR1 and FPR2 to form the major interaction sites (Fig. 2b, c). Residues I344, L348, L353, and F354 of $G_{\alpha i}$ form a hydrophobic

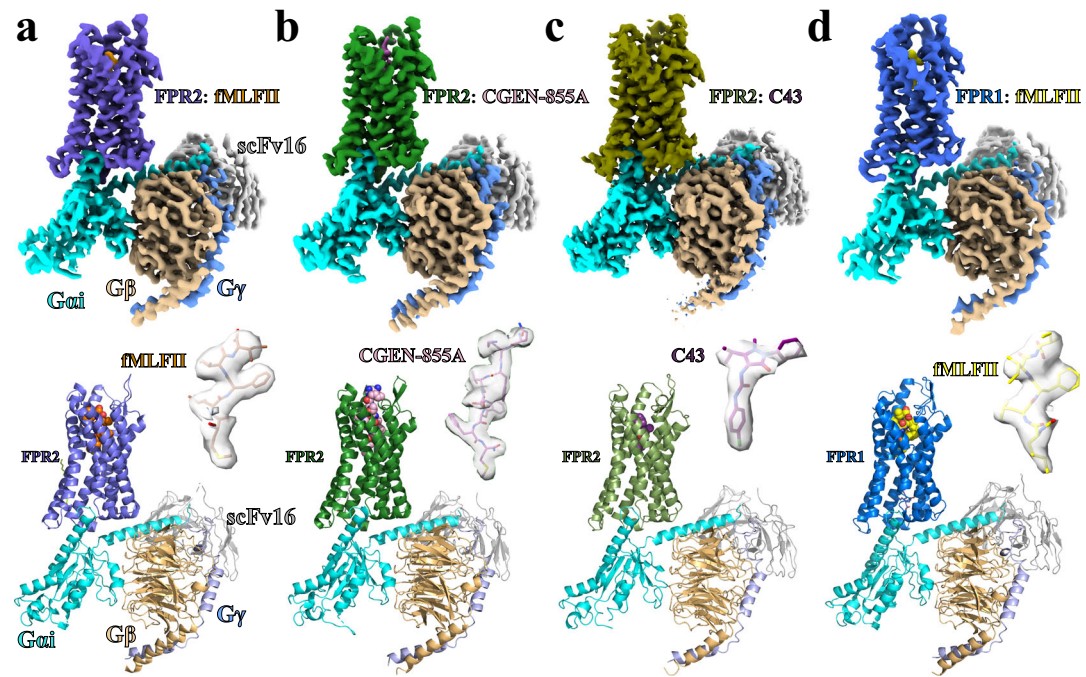

**Fig. 1 Overall structures of the G$_i$-coupled FPR1 and FPR2 complexes. a–d** Upper panels show cryo-EM density maps of the FPR2-G$_i$ complexes with fMLFII (**a**), CGEN-855A (**b**), C43 (**c**), and the FPR1-G$_i$ complex with fMLFII (**d**). Lower panels show structural models of the four complexes and ligand density maps.

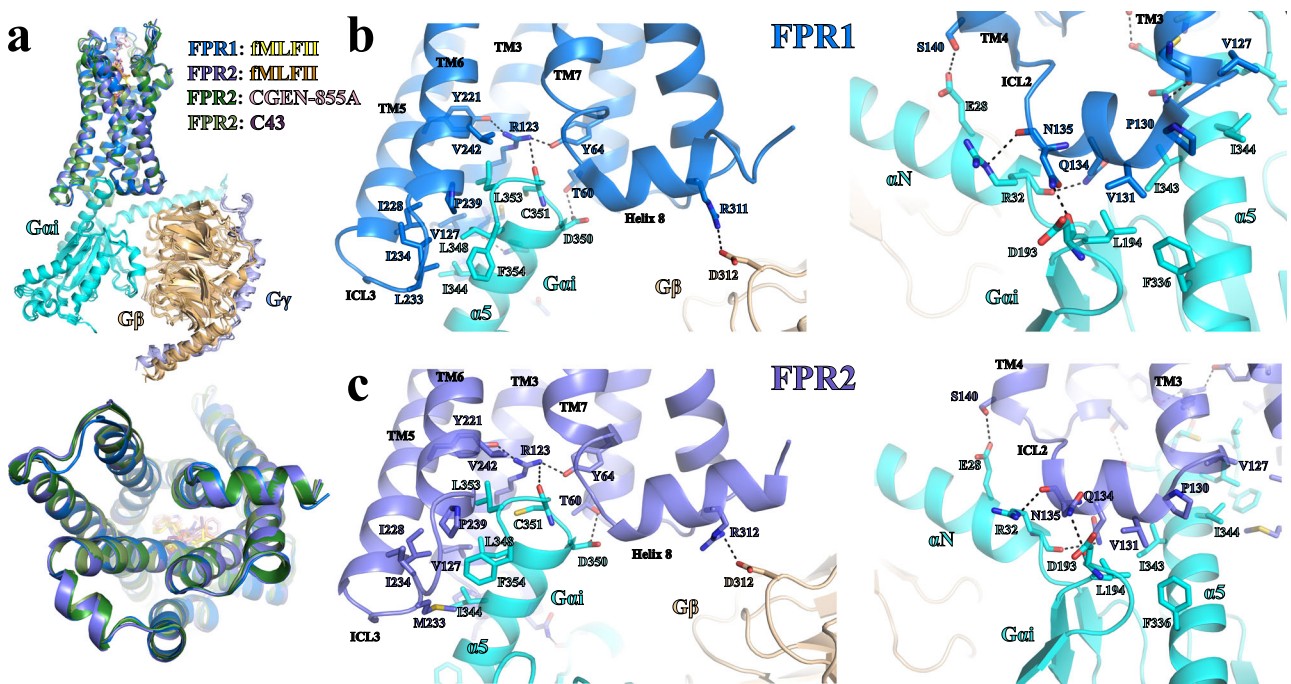

**Fig. 2 Conserved G$_i$-coupling mode for FPR1 and FPR2. a** Structural superimposition of the FPR1- and FPR2-G$_i$ complexes based on the alignment of FPR1 and FPR2. The lower panel shows the structural alignment of the cytoplasmic regions of FPR1 and FPR2 in four structures. **b, c** Detailed interactions between the two FPRs and G$_i$ viewed from two angles. FPR1 and FPR2 are colored in slate and blue, respectively. G$_{\alpha i}$, G$_\beta$, and G$_\gamma$ are colored in cyan, sand, and light blue, respectively. Hydrogen bonds are shown as black dashed lines.

cluster with nearby residues of FPR1 and FPR2 including the ICL3 residue M233 in FPR2 and L233 in FPR1 (Fig. 2b, c). In both receptors, R123$^{3.50}$ (superscripts represent Ballesteros-Weinstein numbering[42]) in the conserved DR$^{3.50}$Y/F/C motif mediates a hydrogen bond network with the side chains of

receptor residues Y64$^{2.43}$ and Y221$^{5.58}$ and the main-chain carbonyl of C351 of G$_\alpha$ (Fig. 2b, c). Besides the cytoplasmic cavities, ICL2s in FPR1 and FPR2 is also involved in direct interactions with G$_{\alpha i}$. The ICL2 residues P130 and V131 in both receptors form another hydrophobic cluster with G$_{\alpha i}$ residues

L194, F336, I343, and I344, while polar residues Q134, N135, and T138 in ICL2 form multiple hydrogen-bonding interactions with αN of $G_{\alpha i}$ (Fig. 2b, c). In addition, both receptors engage in direct polar interactions with $G_{\beta}$ (Fig. 2b, c).

**Recognition of formylpeptides by FPR1 and FPR2 and receptor activation.** In our structures, the formylpeptide fMLFII inserts into the binding pockets of FPR1 and FPR2 in an extended conformation similar to that of the non-formylpeptide WKYMVm[40]. The N-formyl methionine residue of fMLFII is buried inside the 7-TM buddle, overlapping with the C-terminal d-methionine residue of WKYMVm (Supplementary Fig. 5a–d). Such an 'N-terminus-inside' binding pose of fMLFII is consistent with our previous docking study[40]. The side chain of the N-formyl methionine inserts into a narrow chamber at the bottom region of the ligand-binding pocket in both structures (Fig. 3a), forming a hydrophobic cluster with highly conserved residues L109[3.36], F110[3.37], V113[3.40], and W254[6.48] in FPR1 and FPR2 (Fig. 3b, c). Among these residues, V[3.40] together with P[5.50] and F[6.44] constitute a conserved 'core triad' motif that has been suggested to participate in the allosteric conformational prorogation in the activation of several Class A GPCRs such as the β2-adrenergic receptor (β2AR)[43] and μ-opioid receptor (MOR)[44]. Rearrangement of W[6.48] and F[6.44] has been observed in the activation of many Class A GPCRs, which is associated with conformational changes at the cytoplasmic regions including the outward displacement of TM6, a hallmark of GPCR activation[45–48]. Therefore, we propose that the narrow chamber in FPR1 and FPR2 accommodating the side chain of the N-formyl methionine functions as an 'activation chamber' of FPRs (Fig. 3b), where formylpeptides directly interact with and cause conformational changes of the core region of the 7-TMs to activate the receptors.

Previously, a crystal structure of FPR2 alone with WKYMVm was reported[31]. Alignment of this structure with the structures of Gi-coupled FPR2 bound to WKYMVm and fMLFII indicated very similar binding poses of the peptide ligands (Supplementary Fig. 5e). Unexpectedly, the conformation of FPR2 alone with WMYMVm highly resembles the fully active conformation of Gi-coupled FPR2, especially for the transmembrane region including TM5, TM6, and TM7, suggesting that the peptide agonist alone could stabilize or induce the active conformation of FPR2 (Supplementary Fig. 5e). In contrast, a crystal structure of β2AR bound to an irreversible agonist revealed an inactive conformation of the receptor[49]. Nevertheless, ICL3 showed a different conformation in the Gi-coupled FPR2 (Supplementary Fig. 5e), which is likely caused by the direct interactions between ICL3 and Gi.

At the mouth of the activation chamber, fMLFII engages in extensive polar interactions with three conserved residues of FPR1 and FPR2, D106[3.33], R201[5.38], and R205[5.42] (Fig. 3b, c). In FPR2, the N-formyl group of fMLFII is positioned right below the side chain of R201[5.38], forming hydrogen bonds with R201[5.38]. R201[5.38] also forms a hydrogen bond with the main-chain carbonyl group of Leu2 of fMLFII and a salt bridge with D106[3.33], which in turn forms salt bridges with the main-chain amine groups of fMet1 and Leu2 of fMLFII (Fig. 3b, c). In addition, R205[5.42] engage in hydrogen-bonding interactions with the main-chain carbonyl groups of fMet1 and Leu2 of fMLFII (Fig. 3b, c). In the structure of fMLFII-bound FPR1, similar polar interactions among fMLFII and D106[3.33], R201[5.38], and R205[5.42] of FPR1 are also observed (Fig. 3b, c). It is likely that those polar interactions involving the N-formyl group are important for holding the formylpeptide at the mouth of the activation chamber to allow the side chain of the formylated methionine to insert into the narrow activation chamber properly. To support this hypothesis,

we performed mutagenesis studies on individual mutations of D106[3.33], R201[5.38], and R205[5.42] to alanine in FPR1, which led to compromised agonistic potency of fMLF, the prototypical formylpeptide agonist of FPR1 (Fig. 3d, Supplementary Fig. 6a, b). For FPR2, previous studies from us and others showed that individual mutations of D106[3.33], R201[5.38], and R205[5.42] could even result in undetectable agonistic activities of formylpeptides[31,40] (Supplementary Fig. 6c). Together, these data suggested an important role of this triad of polar residues in the formylpeptide recognition by FPRs.

**Structural differences in the ligand-binding pockets of FPR1 and FPR2.** Despite a high similarity, there are notable differences in the ligand-binding pockets of FPR1 and FPR2, which result in slightly different ligand-binding modes of the formylpeptide ligand (Supplementary Fig. 5a, b). At the mouth of the activation chamber, F257[6.51] of FPR2 is replaced by Y257[6.51] in FPR1, which forms an additional hydrogen bond with the main-chain carbonyl of fMet1 of fMLFII (Fig. 3b, c). The side chain of Leu2 of fMLFII is surrounded by aromatic residues F81[2.60], F102[3.29], V105[3.32], and F291[7.43] in FPR1 to form extensive hydrophobic interactions, while it only forms hydrophobic interactions with L81[2.60], V105[3.32], and F292[7.43] in FPR2 since F102[3.29] in FPR1 is replaced by H102[3.29] in FPR2 (Fig. 3b, c). Those additional interactions between fMLFII with FPR1 explain the ~100-fold higher potency of fMLFII in activating FPR1 than does FPR2 as reported in a previous study[32]. Consistently, our mutagenesis data showed that the mutations of Y257[6.51] and F102[3.29] in FPR1 to their counterparts in FPR2 resulted in reduced potency of fMLF in activating FPR1, indicating lowered potency (Fig. 4a, Supplementary Fig. 6a, b). In addition, Phe3-Ile5 of fMLFII forms hydrophobic interactions with different sets of residues from FPR1 and FPR2 (Fig. 3b, c).

Previous studies from our group[40] and others[32] predicted opposite charge distributions of the ligand-binding pockets in FPR1 and FPR2, offering another structural explanation for the preference of FPR2 for long formylpeptides over short formylpeptides[40]. For example, fMLF was indicated to be about 1000-fold less potent (in terms of $EC_{50}$) in inducing FPR2 signaling than does FPR1[32]. Indeed, the electrostatic potential maps calculated based on our structures confirm the negatively and positively charged binding pockets in FPR2 and FPR1, respectively (Fig. 4b). It was suggested that the negative electrostatic potential provided by the FPR2 residue D281[7.32], which corresponds to G280[7.32] in FPR1 (Fig. 4c), disfavors the binding of the short formylpeptide fMLF in FPR2 (Supplementary Fig. 6c)[32]. In FPR1, our structural analysis suggests that the side chain of R84[2.63], which corresponds to S84[2.63] in FPR2, sticks towards the peptide-binding site and contributes to the positive charge potential of the FPR1 ligand-binding pocket (Fig. 4c). The electrostatic potential-switching mutation R84D reduces the potency of fMLF in activating FPR1 (Fig. 4a, Supplementary Fig. 6b), indicating an important role of the positive potential in the recognition of short formylpeptides by FPR1.

The extracellular loops of FPR1 and FPR2 also show large structural differences (Fig. 4c). Compared to them in FPR2, the extracellular loops ECL2 and ECL3 in FPR1 are positioned closer to each other, resulting in a less open extracellular region (Fig. 4c, d). The narrow extracellular opening of FPR1 may restrict the access of long formylpeptides, therefore at least partly accounting for the preference of FPR1 against long formylpeptides.

**Molecular basis for the action of peptide and non-peptide FPR2 agonists.** To investigate how FPR2 recognizes diverse

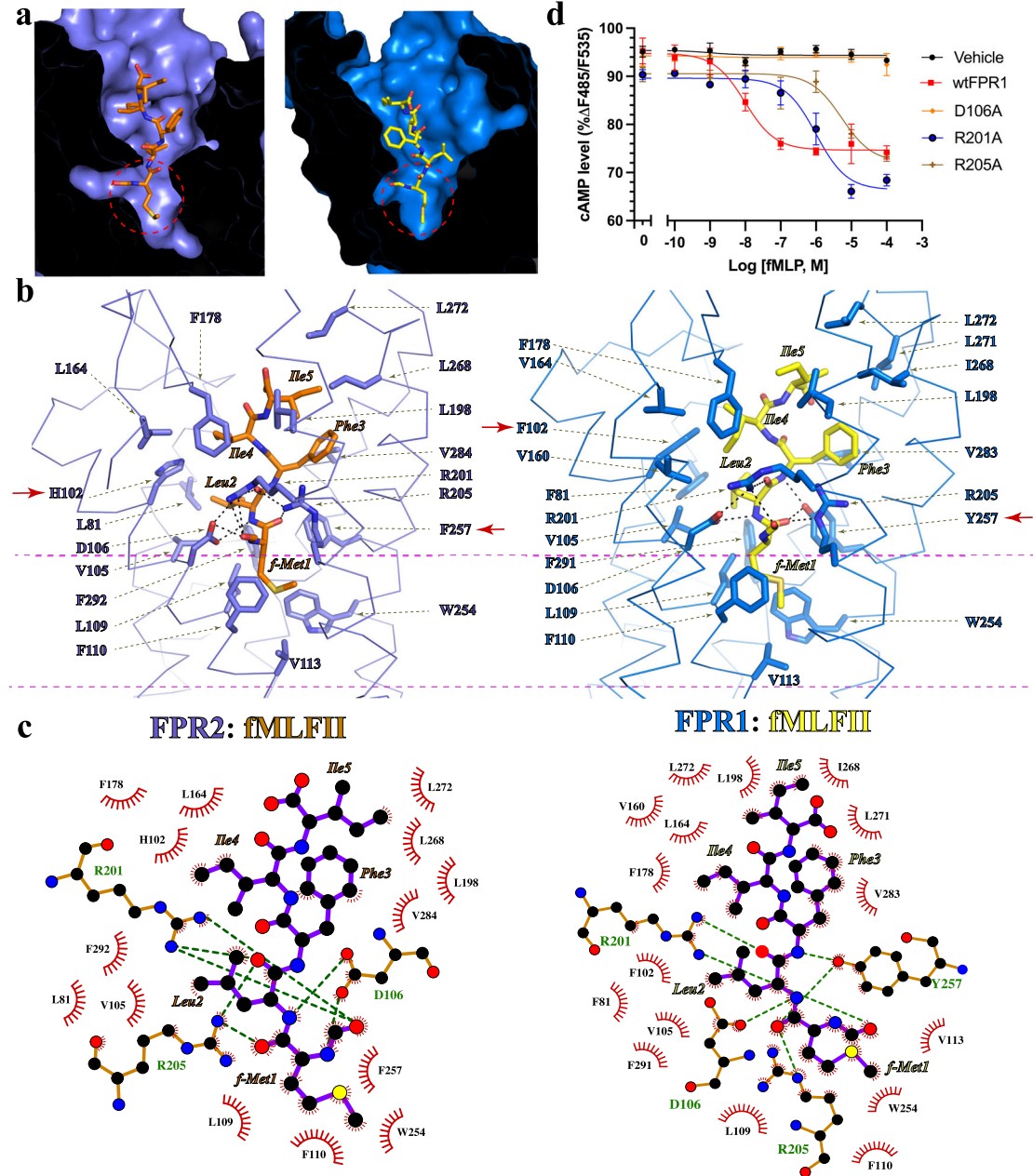

**Fig. 3 Binding of fMLFII to FPR1 and FPR2. a** Overall binding modes of fMLFII in FPR1 (blue) and FPR2 (slate). The N-terminal formyl methionine residue of fMLFII is circled. **b** Details of binding pockets of fMLFII in FPR2 (left) and FPR1 (right). Red arrows point to non-conserved residues H102 and F257 in FPR2 and F102 and Y257 in FPR1. Polar interactions are shown as black dashed lines. The activation chamber is shown between the two purple dashed lines. **c** Ligplot schematic representation of fMLFII interactions with FPR2 (left) and FPR2 (right). The carbon, nitrogen, oxygen, and sulfur atoms are colored in black, blue, red, and yellow, respectively. Residues in FPR1 and FPR2 that form polar interactions with fMLFII are shown with green labels. Polar interactions are shown as green dashed lines. The ligand is shown as purple sticks. **d** Dose-dependent action of fMLFII on wide type FPR1 (wtFPR1) and mutants. Agonist-induced FPR1 signaling was measured by cAMP accumulation assay. Each data point represents mean ± S.D. Three independent assays were performed. Source data are provided as a Source Data file.

agonists, we also determined cryo-EM structures of FPR2 with CGEN-855A and C43 (Fig. 1, Supplementary Fig. 3). CGEN-885A is a FPR2-selective peptide agonist with a carboxy-terminal amidated methionine residue. Although CGEN-885A and WKYMVm share little sequence similarity, they both bind to FPR2 in a 'C-terminus-inside' mode with similar binding poses (Supplementary Fig. 7). The three C-terminal residues of CGEN-885A and WKYMVm can be well superimposed (Supplementary Fig. 7). The side chain of the C-terminal Met21 of CGEN-885A sticks into the activation chamber to form hydrophobic

interactions with L109$^{3.36}$, F110$^{3.37}$, V113$^{3.40}$, and W254$^{6.48}$ of FPR2 (Fig. 5a, b). Two other hydrophobic clusters are also observed to contribute to the binding of CGEN-885A. One is formed among side chains of Phe20 of CGEN-885A and L81$^{2.62}$, H102$^{3.29}$, and F292$^{7.43}$ of FPR2, and the other is formed among side chains of Trp21 and Phe16 of CGEN-885A and F178$^{ECL2}$, L198$^{5.35}$, L268$^{6.62}$, L272$^{ECL3}$, and V284$^{7.35}$ of FPR2 (Fig. 5a, b). Besides these hydrophobic interactions, CGEN-885A also forms multiple hydrogen bonds and salt bridges with the three polar residues D106$^{3.33}$, R201$^{5.38}$, and R205$^{5.42}$ of FPR2 (Fig. 5a, b).

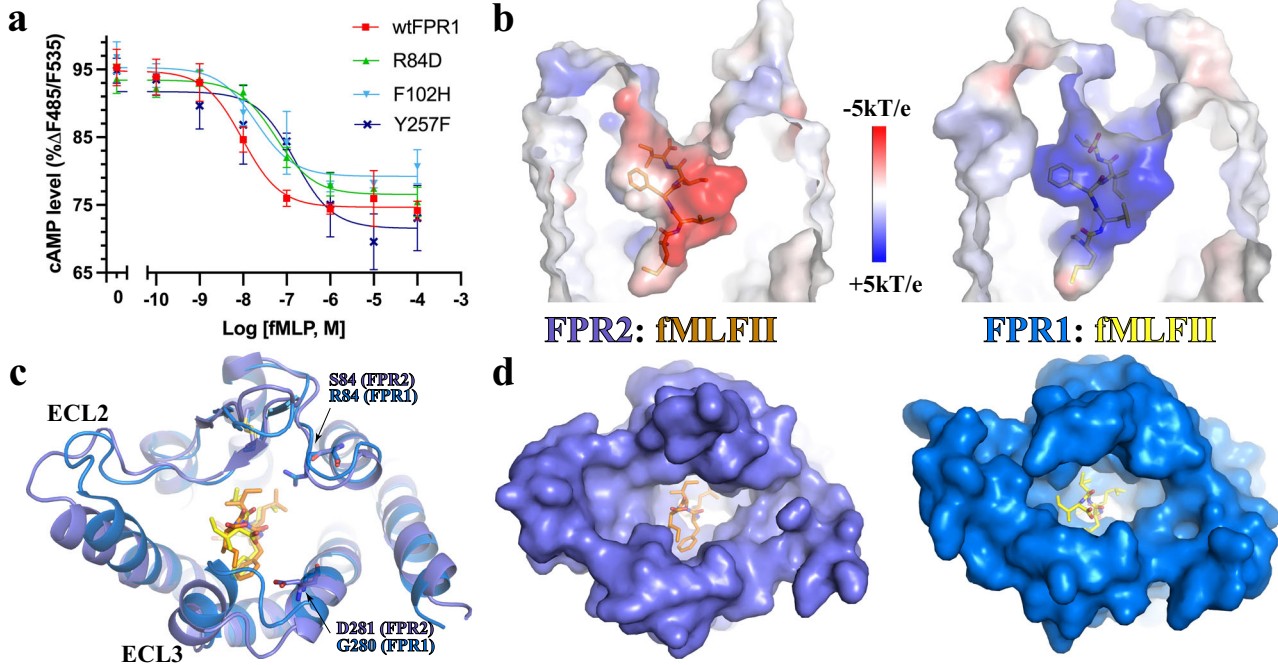

**Fig. 4 Structural differences between FPR1 and FPR2. a** Dose-dependent action of fMLFII on wide type FPR1 (wtFPR1) and FPR1 with mutations of three non-conserved residues to their counterparts in FPR2. Agonist-induced FPR1 signaling was measured by cAMP accumulation assay. Each data point represents mean ± S.D. Three independent assays were performed. Source data are provided as a Source Data file. **b** Charge distributions of fMLFII-binding pockets in FPR1 and FPR2. Overall binding modes of fMLFII in FPR1 (blue) and FPR2 (slate). **c** Alignment of the extracellular regions of FPR1 and FPR2 showing different conformations of ECL2 and ECL3. Two non-conserved residues, S84 and D283 in FPR2 (slate) and R84 and G280 in FPR1 (blue), are shown as sticks. **d** Structural comparison of the extracellular surfaces of FPR1 and FPR2. fMLFII is colored orange in FPR2 and yellow in FPR1. ECL extracellular loop.

To investigate critical structural determinants for the binding of CGEN-885A, we measured the CGEN-885A-induced signaling of FPR2 with mutations of residues in the CGEN-885A-binding pocket. To our surprise, most of the mutations we tested with the exception of D106[3.33]A showed little effect on the potency of CGEN-885A in inducing FPR2 signaling (Fig. 5c, Supplementary Fig. 6a, b), contrasting with the much compromised agonistic activities of formylpeptides or WKYMVm caused by either R201[5.38]A or R205[5.42]A mutation[31,32,40]. Considering the negatively charged environment of the ligand-binding pocket in FPR2 (Fig. 4b), it is possible that the binding of CGEN-885A to FPR2 is largely driven by the opposite charge attraction between FPR2, especially D106[3.33], and the C-terminal amine group of CGEN-885A. Interestingly, structural alignment of CGEN-885A-bound FPR2 and FPR1 indicated that the upper region of the ligand-binding pocket of FPR1 is too narrow for CGEN-885A and there would be a severe clash of the peptide agonist with ECL2 and ECL3 of FPR1 (Fig. 5d), explaining the selectivity of CGEN-885A for FPR2 over FPR1[33].

As a non-peptide synthetic FPR2 agonist, C43 sits at the bottom region of the ligand-binding pocket. The urea moiety exhibits a W-shaped extended conformation[34] to allow the chlorophenyl group of C43 to stick into the activation chamber, where it forms a large hydrophobic cluster with FPR2 residues L109[3.36], F110[3.37], V113[3.40], W254[6.48], F257[6.51], and F292[7.43] (Fig. 6a, b). At the mouth region of the chamber, the two carbonyl groups of C43 in the urea and pyrazolone moieties, respectively, engage in multiple hydrogen bonds with R201[5.38] and R205[5.42] of FPR2, while the urea amine group forms a salt bridge with D106[3.33] of FPR2 (Fig. 6a, b). In addition, the methylethyl group attached to the pyrazolone group of C43 forms hydrophobic interactions with

L81[2.62], H102[3.29], V284[7.35], and F292[7.43] of FPR2 (Fig. 6a, b). Individual mutations of R201[5.38] and R205[5.42] at the mouth region or F110[3.37] in the activation chamber could lead to nearly undetectable agonistic action of C43 (Fig. 6c), suggesting important roles of these residues in the binding of C43.

Structural comparison of FPR2 bound to C43 and peptide agonists indicates that the chlorophenyl group of C43 attached to the urea moiety overlaps with the terminal methionine residues in three peptide agonists (Fig. 6d), which implies a conserved receptor activation mechanism for both peptide and non-peptide FPR2 agonists. Similar to peptide agonists, C43 causes conformational changes of residues in the activation chamber including V113[3.40] and W254[6.48] through direct interactions to activate the receptor. Interestingly, previous structure-activity relationship (SAR) studies on C43 and related compounds showed that the size and position of substituents in the urea phenyl group were important for the agonist activity[34]. For a series of C43-related compounds, replacing the para-chloride group with bigger halides or smaller groups led to increased or decreased potency, respectively[34]. In fact, one compound with an unsubstituted urea phenyl group showed little measurable activity[34]. Alternating the para-position of the chloride group in C43 resulted in a significant loss of ligand potency[34]. Collectively, these results suggest that for C43 and its derivatives, the chemical groups occupying the activation chamber of FPR2 need to be of sufficient size or length in order to induce conformational changes of critical residues in the chamber to activate the receptor.

**Discussion**
FPRs are unique GPCRs with functional promiscuity. Structural comparison analysis of the fMLFII-FPR1-Gᵢ complex and the

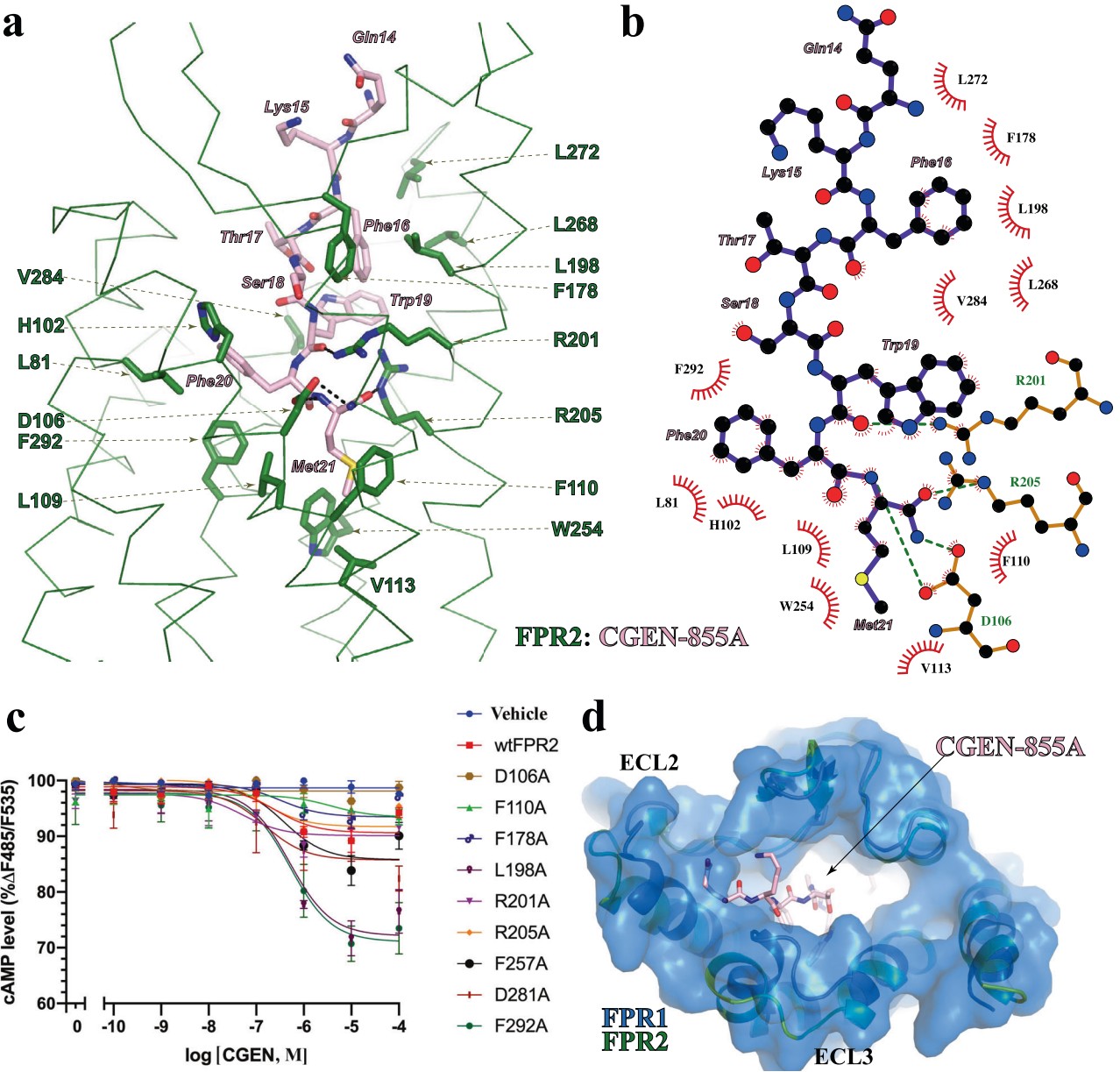

**Fig. 5 Binding of CGEN-855A to FPR2. a** Details of CGEN-855A-binding pocket. **b** Ligplot schematic representation of CGEEN-855A interactions with FPR2. The carbon, nitrogen, oxygen, and sulfur atoms are colored in black, blue, red, and yellow, respectively. The ligand is shown as purple sticks. Residues in FPR2 that form polar interactions with CGEEN-855A are shown with green labels. Polar interactions are shown as green dashed lines. **c** Dose-dependent action of CGEN-855A on wide type FPR2 (wtFPR2) and mutants measured by cAMP accumulation assay. Each data point represents mean ± S.D. Three independent assays were performed. **d** Structural alignment of the CGEN-855A-bound FPR2 to FPR1 showing clash of CGEN-855A with the extracellular region of FPR1. Source data are provided as a Source Data file.

FPR2-$G_i$ complexes with four agonists, WKYMVm, fMLFII, CGEN-885A, and C43, revealed interesting features of agonist binding to these two FPRs. In FPR2, all four agonists sample the narrow activation chamber at the bottom of the ligand-binding pocket formed by conserved residues (Fig. 6e). The side chains of either the N-terminal or the C-terminal methionine residue of three peptide agonists and the chlorophenyl group of C43 can be well-aligned in the activation chamber, while all ligands form extensive polar interactions with three conserved polar residues $D106^{3.33}$, $R201^{5.38}$, and $R205^{5.42}$ at the mouth region of the activation chamber. The same features can be observed in the binding of fMLFII to FPR1. Therefore, we propose that most FPR agonists share a similar receptor activation mechanism, where they occupy the conserved activation chamber in the receptor to cause

conformational changes of residues in this chamber to activate the receptor (Fig. 6e). Indeed, $W^{6.48}$ and $V^{3.40}$ in the activation chamber together with nearby residues $P^{5.50}$ and $F^{6.44}$ form critical core motifs in the activation of Class A GPCRs that link extracellular agonist binding with cytoplasmic G protein-coupling (Fig. 6e)[46,50]. Structural alignment of active FPR2 with other active Class A GPCRs indicated that while $W^{6.48}$ adopts different conformations, $F^{6.44}$ can be well superimposed (Supplementary Fig. 8). Also, for a particular agonist, the three conserved polar residues $D106^{3.33}$, $R201^{5.38}$, and $R205^{5.42}$ at the mouth region constitute a 'grip' to hold the agonist in the correct position through extensive polar interactions (Fig. 6e). This is to ensure that certain chemical groups in the agonist such as the side chain of a methionine residue in peptide agonists or the chlorophenyl group in C43 can

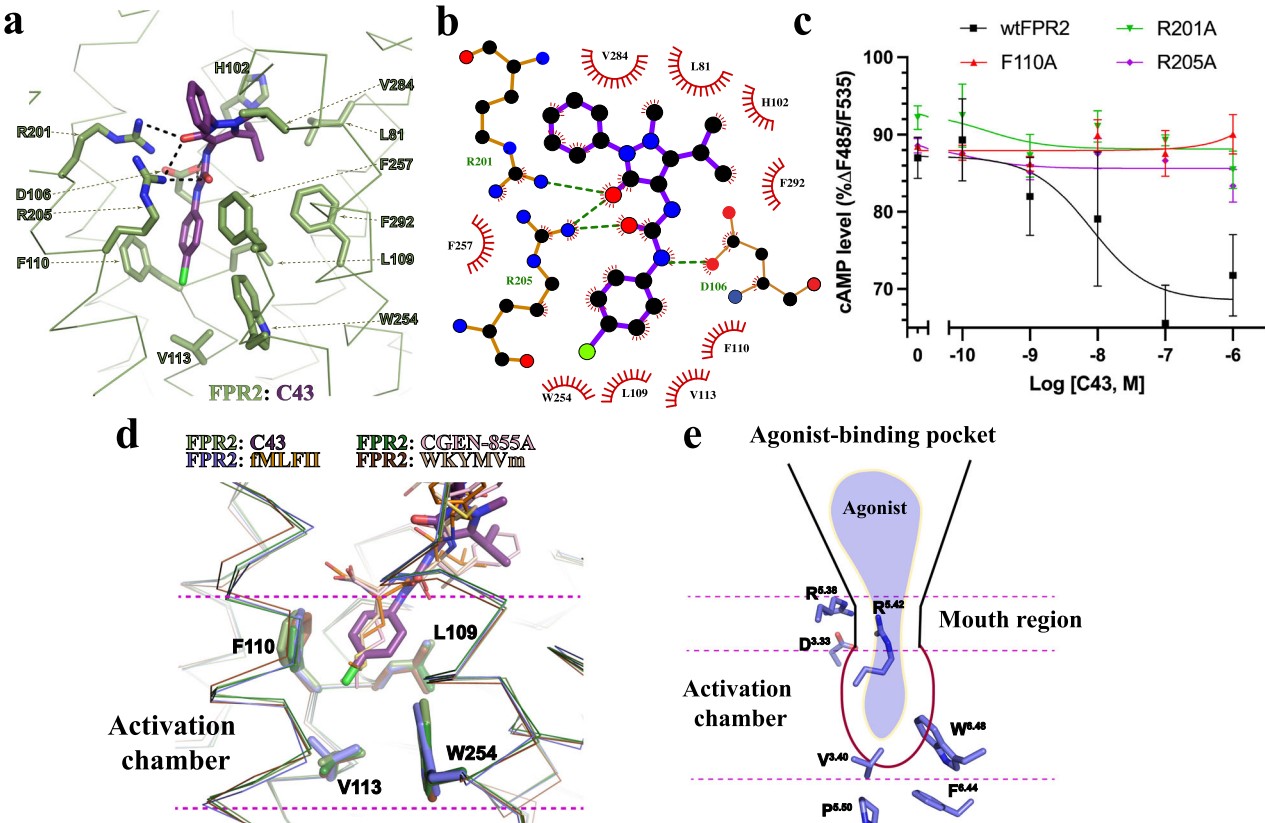

**Fig. 6 Binding of C43 to FPR2. a** Details of C43-binding pocket. FPR2 and C43 are colored in light green and purple, respectively. Polar interactions are shown as black dashed lines. **b** Ligplot schematic representation of C43 interactions with FPR2. The carbon, nitrogen, oxygen, and sulfur atoms are colored in black, blue, red, and yellow, respectively. Residues in FPR2 that form polar interactions with C43 are shown with green labels. Polar interactions are shown as green dashed lines. **c** Dose-dependent action of C43 on wide type FPR2 (wtFPR2) and mutants measured by cAMP accumulation assay. Each data point represents mean ± S.D. Three independent assays were performed. **d** Structural alignment of FPR2 bound to four agonists. The chlorophenyl group of C43 overlaps with the terminal methionine residues in three peptide agonists in the activation chamber. **e** Conserved agonist-binding mechanism for FPR1 and FPR2. For a particular agonist, it sticks into the activation chamber to cause conformational changes of $V^{3.40}$, $W^{6.48}$, $P^{5.50}$, and $F^{6.44}$ to activate receptors. This is facilitated by a polar interaction network with $D^{3.33}$, $R^{5.42}$, and $R^{5.38}$ at the mouth region. Above the mouth region, the vase space of agonist-binding pocket may tolerate a large chemical diversity. Source data are provided as a Source Data file.

insert into the activation chamber in an appropriate conformation to activate receptors. In this scenario, the N-formyl group of formylpeptides may facilitate the formation of extensive polar interaction networks involving $D106^{3.33}$, $R201^{5.38}$, and $R205^{5.42}$ to allow appropriate sampling of the activation chamber, explaining the critical role of N-formylation in the recognition of those peptide ligands by FPRs.

Such a receptor activation mechanism seems unique to FPRs. Structural comparison of $G_i$-coupled FPR1 and FPR2 with other $G_i$-coupled Class A GPCRs including µOR[51], NTSR1[41], chemokine receptor 5 (CCR5)[52], cholecystokinin receptor B (CCKBR)[53], and orexin receptor 2 (OX2R)[54] bound to different peptide agonists indicated that the peptide agonists in those GPCRs are all positioned above W(or Y)[6.48], while fMLFII inserts more deeply into the core regions of FPR1 and FPR2 to reach the activation chamber to activate the receptors (Supplementary Fig. 9). In addition, structural comparison of $G_i$-coupled FPR2 and FPR2 alone suggested that the binding of agonists in the activation chamber is sufficient to induce or stabilize the active conformation of FPR2 for $G_i$-coupling (Supplementary Fig. 5e). This is different from some other Class A GPCRs including β2AR and MOR, for which previous structural and biophysical studies suggested a loose coupling of the extracellular agonist-binding events and receptor conformational changes at the cytoplasmic G protein-coupling region[55,56].

FPR1 and FPR2 have different preferences for formylpeptides, which stem from their structural differences in extracellular regions (Fig. 4). The non-conserved residues $Y257^{6.51}$ and $F102^{3.29}$ in FPR1 form direct interactions with fMLFII, which are absent in the fMLFII-bound FPR2 (Fig. 3b). These additional interactions and the positive electrostatic potential of the ligand-binding pocket of FPR1 result in the higher affinity of the short peptide fMLFII for FPR1 than for FPR2[32]. On the other hand, the restricted opening at the extracellular surface of FPR1 provides an explanation for the selectivity of long peptide agonist CGEN-885A for FPR2 over FPR1 (Fig. 5d). For FPR2, the widely open extracellular surface and the vast space of ligand-binding pocket may tolerate multiple peptide conformations and offer possibilities for diverse peptide and receptor binding interactions above the activation chamber. Therefore, FPR2 can recognize diverse endogenous peptide ligands with different lengths even without N-formylation[2]. This is reminiscent of promiscuous recognition of chemokine agonists by a viral GPCR, US28[57].

Functionally distinct FPR2 agonists have been suggested to induce different conformational states of FPR2[30,58]. However, the structural alignment of the FPR2-$G_i$ complexes with three agonists showed very subtle structural differences for the receptor. This may be due to the coupling of $G_i$, which stabilizes the receptor in one active conformation. Thus, it is difficult to speculate on the specific conformational states of FPR2 associated

with pro-inflammatory and pro-resolving agonists. Previous studies on two FPR2 agonists exhibiting anti-inflammatory or pro-resolving effects, aspirin-triggered 15-epi-lipoxin $A_4$ (ATL) and $Ac_{2-26}$, an N-terminal peptide of ANXA1, showed that both ligands could induce unusual biphasic signaling responses at various concentrations[58,59]. Such results cannot be explained by one specific binding mode of either agonist for FPR2. It is likely that both ligands target multiple binding sites in FPR2 at different concentrations. For the lipid ligands of FPR2, it is tempting to speculate that they occupy similar sites as the palmitic acid molecules modeled in our structures (Supplementary Fig. 4a, b). Interestingly, a similar ligand-binding site close to ICL2 has been identified as the allosteric site for multiple allosteric modulators of C5aR[60], a close phylogenetic neighbor of FPR2. On the other hand, for the small-molecule FPR2 agonist C43, our structure only revealed one orthosteric site overlapping with the binding site of formylpeptides. For this compound and other anti-inflammatory synthetic FPR2 agonists like BMS-986235[38], it is not clear whether the orthosteric binding in FPR2 is sufficient for their anti-inflammatory function. Further studies on the molecular mechanisms of why different FPR2 agonists can induce distinct physiological functions will be important for fully exploiting the therapeutic potential of targeting FPR2 in various diseases.

## Methods

**Construct design**. We use the wild-type human FPR1 and FPR2 for structural studies. Both FPR1 and FPR2 were constructed into the pFastBac vector (ThermoFisher) for expression usage. To facilitate protein expression and purification, the coding sequence of FPR1 (residues 1–333) was fused with an N-terminal FLAG tag followed by a TEV cleavage site, and a C-terminal HIV 3 C protease site-oMBP- MBP- His8 tag. For FPR2, the full-length sequence of FPR2 (residues 1–342) was inserted with an N-terminal FLAG tag followed by β2AR N-terminal region (BN, hereafter) as a fusion protein and a TEV cleavage site, along with a His8 tag at the very C-terminus. The prolactin precursor sequence was added into the N-terminus as a signaling peptide to assist in anchoring FPR1 or FPR2 to the cell membrane and improve their expression. Two dominant-negative mutations, G203A and A326S, were incorporated into the human $G_{\alpha i1}$ ($G_{\alpha i1}$\_2M) by site directed mutagenesis to decrease the binding of GDP/GTP and increase the stability of G protein. Additionally, a His8 tag was cloned onto the N-terminus of $G_\beta$ for two-step purification strategy usage. All the three components of $G_{i1}$ heterotrimer, human $G_{\alpha i1}$\_2M, rat His8- $G_\beta$ and bovine $G_\gamma$, were constructed into the pFastBac vector (ThermoFisher), respectively.

For the expression of scFv16, the coding sequence of scFv16 was fused with a GP67 signaling peptide at the N-terminus and a TEV cleavage site-His8 at the C-terminus, and then cloned into the pFastBac vector (ThermoFisher).

**Expression and purification of the FPRs-Gi signaling complexes**. scFv16 was expressed from Sf9 insect cells as secreted proteins using the baculovirus system (ThermoFisher)[36,40]. To purify the protein, the cell culture supernatant was supplemented with 1 mM $Ni_2SO_4$ and then loaded onto Ni-NTA resins (ThermoFisher). After washing the resins with buffer containing 20 mM Hepes 7.5, 100 mM NaCl, and 50 mM imidazole, scFv16 was eluted in the same buffer with additional 250 mM imidazole. The eluted protein was treated with TEV protease and then dialyzed against 20 mM Hepes pH 7.2, 100 mM NaCl overnight to reduce the concentration of imidazole. After dialysis, the sample was re-loaded onto Ni-NTA resins to remove un-cleaved protein with a His-tag. The protein was further purified by size exclusion chromatography using a HiLoad 16/600 Superdex 200 pg column (GE Healthcare). The protein fractions were pooled, concentrated, and stored at −80 °C.

The baculoviruses of FPR1 or FPR2, $G_{\alpha i1}$\_2M, His8-$G_{\beta 1}$ and $G_{\gamma 2}$ were generated and amplified using the Bac-to-Bac baculovirus expression system (ThermoFisher). The Sf9 cells were cultured in ESF 921 serum-free medium (Expression Systems). When the cell density reached $4 \times 10^6$ cells / mL, we co-expressed the four types of baculoviruses expressing FPR1 or FPR2, $G_{\alpha i1}$\_2M, His8-$G_{\beta 1}$ and $G_{\gamma 2}$ in Sf9 insect cells (Invitrogen) at the ratio of 1:1:1:1. After infection for 48 h, the cells were collected by centrifugation at $1500 \times g$ (ThermoFisher, H12000) for 20 min and kept frozen at −80 °C for complex purification usage.

For the purification of agonists bound FPR2-$G_i$ complexes, cell pellets from 1 l culture were thawed at room temperature and resuspended in low salt buffer containing 20 mM HEPES pH 7.2, 75 mM NaCl, 5 mM $CaCl_2$, 5 mM $MgCl_2$, 0.3 mM TECP, protease inhibitor cocktail (Bimake, 1 mL/ 100 mL suspension). The FPR2-$G_i$ complexes were assembled on the membrane by the addition of peptide ligands 10 μM fMLFII (Genscript Biotech) or 20 μM CGEN-855A

(Genscript Biotech), or synthesized compound ligand 10 μM Compound 43 (MedChemExpress). Half an hour later, the cell suspension was treated with apyrase (25 mU mL$^{-1}$, NEB), followed by incubation for another 1 h at room temperature. Cell membranes in suspension was solubilized directly by adding 0.5% (w/v) lauryl maltose neopentylglycol (LMNG, Anatrace), 0.1% (w/v) cholesteryl hemisuccinate TRIS salt (CHS, Anatrace), 0.025%(w/v) digitonin (Biosynth). After membrane solubilization for 3 h at 4 °C, the solubilized fraction was isolated by centrifugation at $100,000 \times g$ for 45 min and then incubated overnight at 4 °C with pre-equilibrated Nickel-NTA resin (4 mL resin/1 L cell culture). After batch binding, the nickel resin with immobilized protein complex was manually loaded onto a gravity-flow column. The nickel resin was washed with 10 column volumes of 20 mM HEPES, pH 7.2, 100 mM NaCl, 25 mM imidazole, 0.3 mM TCEP, 0.01% LMNG (w/v), 0.002% CHS (w/v), 0.025% digitonin (w/v), 5 μM fMLFII, or 5 μM CGEN-855A or 5 μM Compound 43 and eluted with the same buffer plus 300 mM imidazole. The Ni-NTA eluate was further incubated by batch binding to 2 mL FLAG resin (Smart-Lifesciences) for 2 h at 4 °C. Detergent was exchanged on FLAG resin by two washing steps in 20 mM HEPES, pH 7.2, 100 mM NaCl, 0.3 mM TCEP, 5 μM ligands supplemented with different detergents: first 0.002% LMNG, 0.0004% CHS, 0.05% digitonin, and then 0.05% digitonin for 10 column volumes each. Subsequently, the FLAG resin was resuspended in a detergent buffer containing 20 mM HEPES, pH 7.2, 100 mM NaCl, 0.3 mM TCEP, 5 μM ligands, 0.05% digitonin, and treated with 1.8 mg scFv16 for 1 h at 4 °C. The material bound to FLAG resin was then eluted in detergent buffer containing 20 mM HEPES, pH 7.2, 100 mM NaCl, 0.3 mM TCEP, 5 μM ligands, 0.05% digitonin, 200 μg/μL FLAG peptide.

For the purification of FPR1-fMLFII-$G_i$ complex, cell pellets from 1 L culture were thawed at room temperature for protein purification. All the purification processes are the same as the above for FPR2-fMLFII-Gi complexes with the following exceptions. The first step Ni-NTA eluate was transferred to pre-equilibrated amylose resin (Smart-Lifesciences) and incubated for 3 h at 4 °C. After concentration gradient detergent exchanging, the amylose resin with bound material was treated with HIV 3 C protease (homemade) and 1.8 mg scFv16 for 1 h at room temperature, the released protein was then collected.

The released protein sample was concentrated to 0.5 mL and loaded onto a Superdex 200 10/300 GL increase column (GE Healthcare) pre-equilibrated with buffer containing 20 mM HEPES, pH 7.2, 100 mM NaCl, 0.05% digitonin, 5 μM ligands. Fractions of the monomeric complex were collected and concentrated using centrifugal filter units with a molecular weight cutoff of 100 K Dalton (MilliporeSigma) for electron microscopy experiments.

**Cryo-EM grid preparation and data acquisition**. For cryo-EM grid preparation, 2.7 μL purified FPR2-fMLFII-$G_i$-scFv16 complex at the concentration of 13 mg mL$^{-1}$, FPR2-CGEN-855A-$G_i$-scFv16 complex at the concentration of 13 mg mL$^{-1}$, FPR2-Compound 43-$G_i$-scFv16 complex at the concentration of 16 mg mL$^{-1}$, FPR1-fMLFII-$G_i$-scFv16 complex at the concentration of 14 mg mL$^{-1}$, was applied individually to EM grids (Quantifoil, 300 mesh Au R1.2/1.3) that were glow-discharged for 50 s in a Vitrobot chamber (FEI Vitrobot Mark IV). Protein concentration was determined by absorbance at 280 nm using a Nanodrop 2000 Spectrophotometer (Thermo Fisher Scientific). The Vitrobot chamber was set to 100% humidity at 4 °C. The sample was blotted for 3 s before plunge-freezing into liquid ethane. The prepared cryo-EM grids were stored in liquid nitrogen for screening and data collection usage.

Before data collection, grids were previously screened with a FEI 200 kV Arctica transmission electron microscope (TEM), the promising grids with evenly distributed particles and thin ice layer were transferred to a FEI 300 kV Titan Krios TEM for further data collection.

For the FPR1-fMLFII-$G_i$-scFv16 and FPR2-fMLFII-$G_i$-scFv16 complexes, automatic cryo-EM movie stacks were collected on a FEI Titan Krios microscope operated at 300 kV in Cryo-Electron Microscopy Research Center, Shanghai Institute of Materia Medica, Chinese Academy of Sciences (Shanghai, China). The microscope was operated with a nominal magnification of 81,000× in super-resolution counting mode, corresponding to a pixel size of 0.5355 Å. A total of 4054 movies for the dataset of FPR1-fMLFII-$G_i$-scFv16 complex, and 4579 movies for the dataset of FPR2 -fMLFII-$G_i$-scFv16 complex were collected individually by a Gatan K3 Summit direct electron detector with a Gatan energy filter (operated with a slit width of 20 eV) (GIF) using the SerialEM software. The images were recorded at a dose rate of about 23.3 e/Å2/s with a defocus ranging from −0.5 to −3.0 μm. The total exposure time was 3 s and intermediate frames were recorded in 0.083 s intervals, resulting in a total of 36 frames per micrograph.

For the FPR2-CGEN-855A-$G_i$-scFv16 complex and FPR2-Compound 43-$G_i$-scFv16 complex, automatic cryo-EM movie stacks were collected on a FEI Titan Krios microscope operated at 300 kV in Shuimu BioSciences Ltd. The microscope was equipped with a Gatan Quantum energy filter and a spherical corrector for data collection. The movie stacks were collected automatically using a Gatan K3 direct electron detector with a nominal magnification of ×64,000 in a super-resolution counting model at a pixel size of 0.54 Å. The energy filter was operated with a slit width of 20 eV. Each movie stack was dose-fractioned in 32 frames with a dose of 1.93 electrons per frame and collected within a defocus ranging from −0.5 to −3.0 μm. The total exposure time was 3.2 s. A total of 3162 movies for the dataset of FPR1-CGEN-855A-$G_i$-scFv16 complex, and 3210 movies for the dataset of

FPR2-Compound 43-$G_i$-scFv16 complex were collected, respectively. Data collection was performed using SerialEM with one exposure per hole on the grid squares.

**Image processing and 3D reconstruction.** Movie stacks were subjected to beam-induced motion correction using MotionCor 2[61]. For the datasets of FPR1-fMLFII-$G_i$-scFv16 and FPR2-fMLFII-$G_i$-scFv16 complexes, the movie stack was aligned, dose weighted and binned by 2 to 1.071 Å per pixel. For the datasets of FPR2-CGEN-855A-$G_i$-scFv16 complex and FPR2-Compound 43-$G_i$-scFv16 complex, movie stacks were aligned, dose weighted and binned by 2 to 1.08 Å per pixel. Data processing was performed using RELION-3.1[62]. Contrast transfer function (CTF) parameters were estimated by Ctffind4[63].

For the FPR1-fMLFII-$G_i$-scFv16 and FPR2-fMLFII-$G_i$-scFv16 complexes, the micrographs with the measured resolution worse than 4.0 Å and micrographs imaged within the carbon area of the grid were discarded, generating 2555 micrographs for the FPR1-fMLFII-$G_i$ dataset and 4049 micrographs for the FPR2-fMLFII-$G_i$-scFv16 dataset to do further data processing. The 3D density map of CCK1R-CCK8-$G_i$ (EMDB ID EMD-31387) low-pass filtered to 40 Å was chosen as a reference map for auto-picking and further 3D classification processes. The 2D and 3D classifications were performed on a binned dataset with a pixel size of 2.142 Å. The auto-picking process produced 2,002,737 particles for FPR1-fMLFII-$G_i$ complex and 5,430,289 particles for FPR2-fMLFII-$G_i$ complex, which were subjected to reference-free 2D classifications to discard bad particles. Particles selected from 2D classification were then subjected to several rounds 3D classifications, resulting in two well-defined subsets with 258,740 particles for FPR1-fMLFII-$G_i$ complex and two well-defined subsets with 303,372 particles for FPR2-fMLFII-$G_i$-scFv16 complex. Subsequent 3D refinement, CTF refinement and Bayesian polishing generated a map with a final global resolution of 3.2 Å for FPR1-fMLFII-$G_i$-scFv16 complex and 3.1 Å for FPR2-fMLFII-$G_i$-scFv16 complex. The resolutions were estimated by applying a soft mask around the protein densities with the FSC 0.143 criteria.

For the FPR2-CGEN-855A-$G_i$-scFv16 and FPR2-Compound 43-$G_i$-scFv16 complexes, the micrographs with the measured resolution worse than 4.0 Å and micrographs imaged within the carbon area of the grid were abandoned, generating 2814 micrographs for the FPR2-CGEN-855A-$G_i$-scFv16 dataset and 3051 micrographs for the FPR2-Compound 43-$G_i$-scFv16 dataset to do further data processing. The 3D density map of CCK1R-CCK8-$G_i$ low-pass filtered to 40 Å was chosen as reference map for auto-picking and further 3D classification processes. The 2D and 3D classifications were performed on a binned dataset with a pixel size of 2.16 Å. The auto-picking process produced 3,057,898 particles for the FPR2-CGEN-855A-$G_i$-scFv16 complex and 4,082,666 particles for the FPR2-Compound 43-$G_i$-scFv16 complex, which were subjected to reference-free 2D classifications to discard fuzzy particles. Particles selected from 2D classification were then subjected to several rounds of 3D classifications, resulting in two well-defined subsets with 594,109 particles for FPR2-CGEN-855A-$G_i$-scFv16 complex and two well-defined subsets with 296,082 particles for FPR2-Compound 43-$G_i$-scFv16 complex. Subsequent 3D refinement, CTF refinement and Bayesian polishing generated a map with an indicated global resolution of 2.9 Å for FPR2-CGEN-855A-$G_i$-scFv16 complex and 3.0 Å for FPR2-Compound 43-$G_i$-scFv16 complex at a Fourier shell correlation of 0.143.

The local resolution map was calculated using ResMap[64]. Surface coloring of the density map was performed using UCSF ChimeraX[65].

**Model building, structure refinement, and figure preparation.** The initial model of FPR1 was built by SWISS-MODEL[66]. The structural models of FPR2, $G_i$ and scFv16 in the structure of WKYMVm-FPR2-$G_i$-scFv16 (PDB ID 6OMM) were used as initial templates. All models were first fitted as rigid bodies into the cryo-EM density maps using Chimera[67]. Then, the combined models were refined in Phenix[68] and manually adjusted in Coot[69]. All ligands were manually modeled in Coot and refined in Phenix. The final models were validated by Molprobity[70]. Cryo-EM data collection, image processing and structure refinement statistics are listed in Supplementary Table 1. Figures showing structural models and detailed structural information were prepared by PyMol 2.4.2 (https://pymol.org/2/). Ligplot schematic representation of ligand interreactions with FPR1 and FPR2 was prepared by LigPlot+ v.2.2.4[71]. The maximum distance cutoffs for polar interactions and hydrophobic interactions were set at 3.5 and 4.5 Å, respectively.

**cAMP accumulation assays.** An Epac-based FRET cAMP biosensor H187 was used to measure cellular cAMP levels[72]. The plasmid encoding H187 was transfected into HEK293 cells to make stable cells reporting intracellular cAMP levels. In brief, the H187 stable cells were seeded in poly-D-lysine treated 96-well black clear-bottom plate at 30,000 cells per well. 24 h later, cells were transiently transfected with DNA encoding the N-terminal FLAG-tagged wild-type FPR1, FPR2 or mutants at an amount of 0.1 μg DNA per well. Assays were performed 24 h after transfection. Cells were washed twice with HBSS buffer and incubated at room temperature for 30 min before the addition of forskolin to a final concentration of 1 μM. The intracellular cAMP level was measured using a multimode reader (Spark 20 M, TECAN) measuring the fluorescence emission intensity at 485 nm and 535 nm upon the excitation at 430 nm. 15 min after forskolin incubation, cells were stimulated with agonists at different concentrations and measured for additional 20 min. The FRET signal was calculated as the normalized FRET ratio of fluorescence emission at 535 nm (F535) divided by emission at 485 nm (F485). The $EC_{50}$ was determined with the FRET ratio at the timepoint of 1600 s. The data analysis was performed using the nonlinear regression fit logIC50 mode in GraphPad Prism.

Receptor cell surface expression was determined by measuring the binding of Alexa Flour 647 labeled anti-FLAG M1 antibody (A647-M1, homemade) to the cell surface. Briefly, transfected cells in the 96-well plate were washed with HBSS buffer and incubated with 1 μg/ml A647-M1 in dark for 30 min at room temperature. The cells were washed twice with HBSS buffer and then measured for the Alexa Flour 647 fluorescence on the multimode reader (Spark 20 M, TECAN). The comparable protein expression levels were represented by the fluorescence values of each well.

**Reporting summary.** Further information on research design is available in the Nature Research Reporting Summary linked to this article.

## Data availability
The 3D cryo-EM density maps the four FPR signaling complexes have been deposited in the Electron Microscopy Data Bank (EMDB) database under following accession codes: EMD-25727 (FPR1-$G_i$-fMLFII-scFV16 complex), EMD-25729 (FPR2-$G_i$-fMLFII-scFV16 complex), EMD-25728 (FPR1-$G_i$-CGEN-855A-scFv16 complex), and EMD-25726 (FPR2-$G_i$-C43-scFV16 complex). The atomic coordinates of the atomic models have been deposited in the Protein Data Bank (PDB) database under the following accession codes: 7T6T (FPR1-$G_i$-fMLFII-scFV16 complex), 7T6V (FPR2-$G_i$-fMLFII-scFV16 complex), 7T6U (FPR1-$G_i$-CGEN-855A-scFv16 complex), and 7T6S (FPR2-$G_i$-C43-scFV16 complex). The raw data for the main Figs. 3d, 4a, 5c, 6c and Supplementary Fig. 6a, b generated in this study are provided in the Source Data file. Other structural models used in this paper from the PDB database are: the NTSR-$G_i$ complex structures at the canonical state (PDB ID 6OS9) and non-canonical state (PDB ID 6OSA); the WKYMVm-FPR2-$G_i$ structure (PDB ID 6OMM); the WKYMVm-FPR2 crystal structure (PDB ID 6LW5); the active structures of μ-opioid receptor (MOR, PDB ID 6DDE), neurotensin receptor 1 (NTSR1, PDB ID 6OS9), and $β_2$-adrenergic receptor (β2AR, PDB ID 3SN6); the peptide-bound structures of chemokine receptor 5 (CCR5, PDB ID 7F1Q), cholecystokinin receptor B (CCKBR, PDB ID 7F8V), and orexin receptor 2 (OX2R, PDB ID 7L1U). Source data are provided with this paper.

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

## Acknowledgements

The cryo-EM data were collected at the Cryo-Electron Microscopy Research Center, Shanghai Institute of Materia Medica (SIMM), and Shuimu BioSciences Ltd (Beijing China). We thank the staff at the SIMM Cryo-Electron Microscopy Research Center and the Shuimu BioSciences Ltd. for their technical support. We thank Dr. Kees Jalink at the Netherlands Cancer Institute for providing us the H187 plasmid. This work was supported by the Ministry of Science and Technology (China) grants (2018YFA0507002 to H.E.X.), the Shanghai Municipal Science and Technology Major Project (2019SHZDZX02 to H.E.X.), the CAS Strategic Priority Research Program (XDB08020303 to H.E.X.), the

Special Research Assistant Project of Chinese Academy of Sciences E1G707R078 (to Y.Z.), and the National Institutes of Health (NIH) grants 1R03TR003306-01 and 1R35GM128641 (to C.Z.).

## Author contributions

Y.Z. designed and screened the expression constructs of FPR1 and FPR2, optimized the protein complexes purification conditions, prepared protein samples of FPR1-fMLFII-G$_i$-scFv16 and FPR2-fMLFII-G$_i$-scFv16 complexes for cryo-EM data collection, prepared and screened the cryo-EM grids. J.G. prepared protein samples of FPR2-CGEN-855A-G$_i$-scFv16 and FPR2-Compound 43-G$_i$-scFv16 complexes for cryo-EM data collection, determined the structures of FPR2-fMLFII-G$_i$-scFv16 and FPR2-Compound 43-G$_i$-scFv16 complexes. Y.Z. performed data acquisition and structure determination of FPR1-fMLFII-G$_i$-scFv16 and FPR2-CGEN-855A-G$_i$-scFv16 complexes, participated in the data processing of FPR2-Compound 43-G$_i$-scFv16 complexes. Y.W. and W.L. assisted in protein sample preparation. L.W. performed cAMP assays and assisted in protein expression and purification. L.W., D.S., and C.Z. modeled and refined all structures. C.Z. and H.E.X. jointly supervised the research. C.Z. and L.W. wrote the manuscript with help from Y.Z. and H.E.X.

## Competing interests

C.Z. serves as a consultant for Biogen. The remaining authors declare no competing interests.
