## [Peer Review File · Nature Communications]

REVIEWER COMMENTS

Reviewer #1 (Remarks to the Author):

In the manuscript "Molecular recognition of formylpeptides and diverse agonists..." by Zhuang et al" the authors present several new structures of the FRP1 (1 structure) and FRP2 (3 structures) GPCRs. The resolution of the structures appears to be sufficient to place the ligands reliably. A main selling point of this manuscript is that functionally distinct ligands act on the FRP2 at different regions to induce-ligand specific conformational changes. To some, very limited end the authors have achieved that, and supported the work with a limited amount of mutagenesis work; however, I believe this manuscript has been rather hastily put together. Therefore, there are quite a few inconsistencies in the manuscript, and more importantly, general readers of the GPCR field might have a hard time following the narrative of this manuscript. Some revision in this respect is required.

Comments.

1) The claim of functionally distinct ligands acting on FRP2 that bind to different regions of the receptor to induce specific-ligand conformations is interesting, however, there is no data presented in this manuscript to back the claim that the ligands used in this study are functionally distinct. Similarly, the cryo-EM structures are all very similar. Is this because the chosen ligands behave quite similarly with these receptors, or G protein? Has this been tested pharmacologically using the constructs used in this study? This is a major limitation of the study. As otherwise, these are just 4 more class A GPCR structures.

2) In the discussion of G protein coupling the authors refer to figure 2b and 2c. There is no figure 2c, and the figure 2 as presented doesn't really seem to fit the narrative the authors are trying to convey.

3) There appear to be lots of previous mutagenesis data on these receptors, as well as some new mutagenesis data. It would be useful to have these listed in a supplemental table.

4) Figure 3, it is difficult to easily identify the residues being discussed in the manuscript and in figure 3. Perhaps add something differentiate the figure labels would make this easier?

5) Regarding figure 4C, with respect to the different ECL2 conformations, are there any structural clues to why these differ? Are they specific to fMLFII or are they also observed in the other FRP2 structures?

6) For CGEN-855A, are there any interactions that were observed in the "upper region" that could be mutated to support claims that region is likely more important.

7) Figure 6D: W6.48 is listed as F6.48 in the text. Also how do the positions of these core Class A GPCR activation residues compare to other class A receptors?

Reviewer #2 (Remarks to the Author):

The manuscript from Zhuang et al reports the structure information of the formylpeptide receptors (FPRs) and their signalling complexes with Gi determined by single particle cryo-EM. The cryo-EM structures of FPR1 and FPR2 in complex with G protein i heterotrimer with different ligands have provided a great insight into the ligand specificity of formylpeptide receptors.

General remarks:

(1) The manuscript is well written and easy to read.

(2) The science part in the manuscript is comprehensive, including not only the structure

determination but also the functional assays.

(3) The authors intensively wrote about the ligand binding pockets in FPR1 and FPR2, with thorough analysis to decipher the selectivity for formylpeptides. However, the authors did not grab the chance to compare FPRs with other peptide receptors (to be simplified, Gi-coupled peptide receptors) where the mechanism for selecting ligands can be discussed not only for FPRs but also for other types of peptide receptors.

(4) Fig. 2c: In the main text, Fig. 2c is mentioned several times in the 2nd paragraph of RESULT section, but there is no Fig. 2c in the manuscript. Please update the figure and the main text accordingly.

(5) Fig. 3b, 5a and 6a: It is important that these figures show the original result for the ligand binding pocket, but I find these figures difficult for readers to trace the ligand-residue contact. Please could you add another sub-figure using LigPlot (<https://www.ebi.ac.uk/thornton-srv/software/LIGPLOT/>) or similar art to illustrate the contact between the ligand and the receptor.

(6) The contact between Gi alpha and FPRs is not addressed very much in detail and seems suppressed by the authors for this topic. Even under the suppressed manner, FPRs-Gi should be at least compared to one of the peptide receptor-Gi complex (for example, neurotensin-Gi). It should also be mentioned that the G protein is in the nucleotide-free state, as this detail only comes out at the METHODS section where Gi alpha contains two mutations for decreasing GDP/GTP binding. Another point is about the state of FPRs-Gi. Are the complexes at the canonical state or the non-canonical state? Does scFv16 promote the complex to be at one state more than the other?

(7) Following up the previous point: As a crystal structure of FPR2+WKYMVm is available, structure comparison should be performed between this structure and the cryo-EM structures. Two basic questions raised here are (1) if the ligand has the same binding pose; (2) How is the TM5-ICL3-TM6 movement upon G protein binding. One supportive figure and discussion for the two questions should be added.

Minor comments:

(8) When describing protein-ligand or protein-protein contact in residues, please supply the distance threshold for such a contact. It can be addressed in the figure legend or in the METHODS.

(9) RESULT>Conserved Gi-coupled modes fro FPR1 and FPR2>2nd paragraph> last sentence: "In addition, both receptors engage in direct polar interactions with Gbeta (Fig. 2b and c), suggesting a direct role of Gbeta in receptor coupling". What do the authors try to say here? At the receptor-Gbeta interface, what residues does Gbeta use to form contact with FPR1 and FPR2, and are these residues conserved in all the 5 Gbeta subtypes?

(10) RESULT>Recognition of formylpeptides by FPR1 and FPR2 and receptor activation>1st paragraph>4th sentence: "(same numbering in both receptors, superscripts represent Ballesteros-Weinstein numbering)". This content in the parentheses seems redundant as it appears in the previous text already.

(11) METHODS>Construct design: please change "pFastbac" to "pFastBac".

(12) METHODS>Construct design>1st paragraph>7th sentence: "Additionally, a His8 tag was cloned onto the N terminal of Gbeta ...". Please change "N terminal" to "N terminus".

(13) METHODS>Expression and purification of the FPRs-Gi signalling complexes>2nd paragraph: Please unify the art of the "hyphen" between FPR and Gi throughout the whole text.

(14) METHODS>Expression and purification of the FPRs-Gi signalling complexes>2nd paragraph>5th sentence: "After membrane solubilization for 3 hours at 4 degree, the supernatant was isolated by centrifugation at ...". Please change "supernatant" to "solubilized fraction".

(15) METHODS>Expression and purification of the FPRs-Gi signalling complexes>2nd paragraph: Please describe how much Nickel-NTA resin is used for the material from 1 L cell culture.

(16) METHODS>Expression and purification of the FPRs-Gi signalling complexes>last paragraph>last sentence: Please describe what type of concentrator is used for concentrating the protein samples.

(17) METHODS>Cryo-EM grid preparation and data acquisition: Please describe if the EM grids are glow discharged before applying protein samples.

(18) METHODS>Image processing and 3D reconstruction: Please add references for RELION, Ctffind4, ResMap, UCSF Chimera (or ChimeraX if used).

(19) METHODS>Image processing and 3D reconstruction> 2nd last paragraph: The complex samples are labelled sometimes with scFv16 and sometimes without. Please label them properly to avoid confusion.

(20) Extended Data Table 1>Refinement: There is no such a thing called "Model resolution". A model is a model with static coordinates of each atom. Please remove the "Model resolution & FSC threshold" in the Refinement part. Instead, the authors can supply the statistics of Map-Model correlation.

When all the remarks above are updated, I suggest Nature Communication to accept this manuscript and publish this great work from Zhuang et al.

Manuscript ID: NCOMMS-21-34674

Title: **Molecular recognition of formylpeptides and diverse agonists by the formylpeptide receptors FPR1 and FPR2**

We thank all reviewers for their constructive comments. Please see our detailed responses to the comments below. The reviewers' comments are in **blue** font and our responses are in **black** font.

Reviewer #1 (Remarks to the Author):

In the manuscript "Molecular recognition of formylpeptides and diverse agonists..." by Zhuang et al" the authors present several new structures of the FRP1 (1 structure) and FRP2 (3 structures) GPCRs. The resolution of the structures appears to be sufficient to place the ligands reliably. A main selling point of this manuscript is that functionally distinct ligands act on the FRP2 at different regions to induce-ligand specific conformational changes. To some, very limited end the authors have achieved that, and supported the work with a limited amount of mutagenesis work; however, I believe this manuscript has been rather hastily put together. Therefore, there are quite a few inconsistencies in the manuscript, and more importantly, general readers of the GPCR field might have a hard time following the narrative of this manuscript. Some revision in this respect is required.

Comments.

1) The claim of functionally distinct ligands acting on FRP2 that bind to different regions of the receptor to induce specific-ligand conformations is interesting, however, there is no data presented in this manuscript to back the claim that the ligands used in this study are functionally distinct. Similarly, the cryo-EM structures are all very similar. Is this because the chosen ligands behave quite similarly with these receptors, or G protein? Has this been tested pharmacologically using the constructs used in this study? This is a major limitation of the study. As otherwise, these are just 4 more class A GPCR structures.

We thank the reviewer for the positive and constructive comments. Our paper is mainly focused on the mechanism of ligand recognition by FPRs. We used three FPR2 agonists including a formylpeptide, a synthetic non-formylpeptide and a small-molecule agonist in our structural studies mainly for their chemical diversity but not for their potential functional differences. Together with mutagenesis data, our structural analysis indicated distinct receptor interaction profiles of these three agonists but a conserved activation mechanism on FPR2. In addition, we reported the first structure of a FPR1-G_i signaling complex. Our structures of FPR1 and FPR2 with the same formylpeptide revealed conserved and non-conserved features of these two receptors in formylpeptide binding, providing a structural basis for the pattern recognition of formylpeptides. The results also revealed the molecular basis for the different ligand preferences of FPR1 and FPR2 by revealing the differences in the ligand pocket topology and hydrophobic/hydrophilic properties between FPR1 and FPR2.

We didn't intend to prove distinct functional properties of the three FPR2 agonists used in our structural studies. In the first paragraph of the section 'Cryo-EM structure determination and overall structures', we simply summarized previous studies on these ligands including contradictory results on the functional properties of the synthetic compound C43. On the other hand, we performed cAMP assays to show that all these FPR agonists could induce cAMP down

regulation through G_i signaling so they were suitable for assembling G_i -coupled FPR signaling complexes. In the last paragraph of our 'Discussion' section, we stated that we observed very subtle structural differences of FPR2 with three agonists likely due to the coupling of G_i and that it is therefore difficult to speculate on the specific conformational states of FPR2 associated with the pro-inflammatory and pro-resolving action of different FPR2 agonists.

2) In the discussion of G protein coupling the authors refer to figure 2b and 2c. There is no figure 2c, and the figure 2 as presented doesn't really seem to fit the narrative the authors are trying to convey.

We apologize for the misuse of figure. We have included a new **Figure 2** in our revised manuscript showing the details of G_i -coupling.

3) There appear to be lots of previous mutagenesis data on these receptors, as well as some new mutagenesis data. It would be useful to have these listed in a supplemental table.

As suggested by the reviewer, we have listed the results from our study and from previous mutagenesis studies in **Extended Data Figure S6b and c**.

4) Figure 3, it is difficult to easily identify the residues being discussed in the manuscript and in figure 3. Perhaps add something differentiate the figure labels would make this easier?

As suggested by the reviewer, we changed the way of labeling residues to make the figures clearer. We also included additional Ligplot schematic representation of ligand interactions with FPR1 or FPR2 in **Figures 3, 5, and 6**.

5) Regarding figure 4C, with respect to the different ECL2 conformations, are there any structural clues to why these differ? Are they specific to fMLFII or are they also observed in the other FRP2 structures?

The different conformations of ECL2 in FPR1 and FPR2 are likely due to their different amino acid sequences. Please see **Figure R1** below showing the sequence alignment of FPR1 and FPR2 (from our previous publication PMID: 32060286). All three structures of FPR2 bound to different ligands are highly similar to each other. The ECL2 in these FPR2 structures is different from that of FPR1 with fMLFII shown in **Figure 4** of our manuscript.

6) For CGEN-855A, are there any interactions that were observed in the “upper region” that could be mutated to support claims that region is likely more important.

We thank the reviewer for the constructive suggestion. We have performed additional mutagenesis studies to test the effect of mutating residues in the upper region, the mouth region, and the activation chamber on the action of CGEN-855A. To our surprise, most of the mutations we tested except D106A showed little effect on the EC₅₀ values of CGEN-855A in inducing FPR2 signaling (**Figure 5c, Extended Figure S6b**). We therefore revised our discussion of CGEN-855A binding in the second paragraph of section 'Molecular basis for the action of peptide and non-peptide FPR2 agonists' accordingly. In our revised manuscript, we proposed that the binding of CGEN-855A to FPR2 is mainly driven by the opposite charge attraction between FPR2, especially D106, and the C-terminal amine group of CGEN-855A.

7) Figure 6D: W6.48 is listed as F6.48 in the text. Also how do the positions of these core Class A GPCR activation residues compare to other class A receptors?

We apologize for the mistake. It was supposed to be F6.44. We also included a new **Extended Data Figure S8** to show the structural alignment of the four core residues in the active structures of FPR2 and three other representative Class A GPCRs. The result suggested that while W6.48 adopt different conformations in those active GPCRs, F6.44 can be well superimposed. We included such discussion in the 'Discussion' section in our revised manuscript.

Reviewer #2 (Remarks to the Author):

The manuscript from Zhuang et al reports the structure information of the formylpeptide receptors (FPRs) and their signalling complexes with Gi determined by single particle cryo-EM. The cryo-EM structures of FPR1 and FPR2 in complex with G protein i heterotrimer with different ligands have provided a great insight into the ligand specificity of formylpeptide receptors.

General remarks:

(1) The manuscript is well written and easy to read.

We thank the reviewer for the positive comment.

(2) The science part in the manuscript is comprehensive, including not only the structure determination but also the functional assays.

We thank the reviewer for the positive comment.

(3) The authors intensively wrote about the ligand binding pockets in FPR1 and FPR2, with thorough analysis to decipher the selectivity for formylpeptides. However, the authors did not grab the chance to compare FPRs with other peptide receptors (to be simplified, Gi-coupled peptide receptors) where the mechanism for selecting ligands can be discussed not only for FPRs but also for other types of peptide receptors.

As suggested by the reviewer, we have included a new **Extended Data Figure 9** to show the structural comparison of fMLFII-bound FPR1 and FPR2 with other G_i-coupled Class A GPCRs

including μ -opioid receptor (MOR), neurotensin receptor 1 (NTSR1), chemokine receptor 5 (CCR5), cholecystokinin receptor B (CCKBR), and orexin receptor 2 (OX2R) bound to different peptide agonists. Compared to other peptide ligands, which are all positioned above the conserved W(or Y)^{6,48}, fMLFII inserts more deeply into the receptor core to reach the activation chamber. Such a binding mode allows the formylpeptide to directly interact with residues in the activation chamber including W^{6,48} to induce receptor activation. We have included the structural comparison analysis at the end of the first paragraph of the 'Discussion' section in our revised manuscript.

(4) Fig. 2c: In the main text, Fig. 2c is mentioned several times in the 2nd paragraph of RESULT section, but there is no Fig. 2c in the manuscript. Please update the figure and the main text accordingly.

We apologize for the misuse of figure. We have included a new **Figure 2** in our revised manuscript showing the details of Gi-coupling to FPR1 and FPR2.

(5) Fig. 3b, 5a and 6a: It is important that these figures show the original result for the ligand binding pocket, but I find these figures difficult for readers to trace the ligand-residue contact. Please could you add another sub-figure using LigPlot (<https://www.ebi.ac.uk/thornton-srv/software/LIGPLOT/>) or similar art to illustrate the contact between the ligand and the receptor.

As suggested by the reviewer, we have included additional Ligplot schematic representation of ligand interactions with FPR1 or FPR2 in **Figures 3, 5, and 6**. We also changed the way of labeling residues in these figures to make them easier to read.

(6) The contact between Gi alpha and FPRs is not addressed very much in detail and seems suppressed by the authors for this topic. Even under the suppressed manner, FPRs-Gi should be at least compared to one of the peptide receptor-Gi complex (for example, neurotensin-Gi). It should also be mentioned that the G protein is in the nucleotide-free state, as this detail only comes out at the METHODS section where Gi alpha contains two mutations for decreasing GDP/GTP binding. Another point is about the state of FPRs-Gi. Are the complexes at the canonical state or the non-canonical state? Does scFv16 promote the complex to be at one state more than the other?

In our revised manuscript, we now specified in the first paragraph of the section 'Cryo-EM structure determination and overall structures' that our complexes were nucleotide free and we used a dominant negative version of human G α_{i1} with mutations to decrease nucleotide-binding. We also updated our **Figure 2** to include details of G $_i$ -coupling to FPR1 and FPR2 and included a new **Extended Data Figure 4c** to show the comparison of the FPR2-G $_i$ complex with the neurotensin receptor 1 (NTSR1)-Gi complex at the canonical and non-canonical state. The results suggested that our structures represent the canonical state. It is not likely that scFv16 promoted the FPR-G $_i$ complexes to be at one state since scFv16 was also used to get the structures of NTSR-Gi at both canonical and non-canonical states (ref PMID: 31243364).

(7) Following up the previous point: As a crystal structure of FPR2+WKYMVm is available, structure comparison should be performed between this structure and the cryo-EM structures. Two basic questions raised here are (1) if the ligand has the same binding pose; (2) How is the

TM5-ICL3-TM6 movement upon G protein binding. One supportive figure and discussion for the two questions should be added.

As suggested by the reviewer, we have added a new figure panel as **Extended Data Fig. S5e** to show the structural comparison of G_i-coupled FPR2 with two peptide agonists and FPR2 alone with WKYMVm. The results revealed highly similar binding poses of the peptide agonists in all three structures. Also, the conformation of FPR2 in the crystal structure highly resembles the activation conformation of G_i-coupled FPR2 in the cryo-EM structures. The transmembrane regions including TM5, TM6 and TM7 can be well superimposed, suggesting that FPR2 alone with WKYMVm in the crystal structure adopted the fully active conformational state. The ICL3 in G_i-coupled FPR2 adopted a different conformation compared to that in FPR2 alone, which is likely caused by the direct interactions between ICL3 and G_i in G_i-coupled FPR2. We have included the discussion in the second paragraph of section 'Recognition of formylpeptides by FPR1 and FPR2 and receptor activation'.

Minor comments:

(8) When describing protein-ligand or protein-protein contact in residues, please supply the distance threshold for such a contact. It can be addressed in the figure legend or in the METHODS.

We have provided the distance threshold for polar and hydrophobic interactions in the 'Model building, structure refinement, and figure preparation' section of Methods.

(9) RESULT>Conserved Gi-coupled modes fro FPR1 and FPR2>2nd paragraph> last sentence: “In addition, both receptors engage in direct polar interactions with Gbeta (Fig. 2b and c), suggesting a direct role of Gbeta in receptor coupling”. What do the authors try to say here? At the receptor-Gbeta interface, what residues does Gbeta use to form contact with FPR1 and FPR2, and are these residues conserved in all the 5 Gbeta subtypes?

We intended to just describe the observation that both FPR1 and FPR2 directly interact with Gbeta. We have updated **Figure 2** to clearly show those interactions. To avoid potential confusion, we have also revised the sentence as "In addition, both receptors engage in direct polar interactions with G_β".

(10) RESULT>Recognition of formylpeptides by FPR1 and FPR2 and receptor activation>1st paragraph>4th sentence: “(same numbering in both receptors, superscripts represent Ballesteros-Weinstein numbering)”. This content in the parentheses seems redundant as it appears in the previous text already.

We have deleted this redundant statement.

(11) METHODS>Construct design: please change “pFastbac” to “pFastBac”

We have made such changes in our revised manuscript.

(12) METHODS>Construct design>1st paragraph>7th sentence: “Additionally, a His8 tag was cloned onto the N terminal of Gbeta ...”. Please change “N terminal” to “N terminus”.

We have corrected this mistake in our revised manuscript.

(13) METHODS>Expression and purification of the FPRs-Gi signalling complexes>2nd paragraph: Please unify the art of the “hyphen” between FPR and Gi throughout the whole text.

We have revised this part accordingly.

(14) METHODS>Expression and purification of the FPRs-Gi signalling complexes>2nd paragraph>5th sentence: “After membrane solubilization for 3 hours at 4 degree, the supernatant was isolated by centrifugation at ...”. Please change “supernatant” to “solubilized fraction”.

We have revised this part accordingly.

(15) METHODS>Expression and purification of the FPRs-Gi signalling complexes>2nd paragraph: Please describe how much Nickel-NTA resin is used for the material from 1 L cell culture.

We have added '4 mL resin/1 L cell culture' in the sentence.

(16) METHODS>Expression and purification of the FPRs-Gi signalling complexes>last paragraph>last sentence: Please describe what type of concentrator is used for concentrating the protein samples.

We have updated the information of the protein concentrators we used.

(17) METHODS>Cryo-EM grid preparation and data acquisition: Please describe if the EM grids are glow discharged before applying protein samples.

We have updated the information of glow-discharging.

(18) METHODS>Image processing and 3D reconstruction: Please add references for RELION, Ctffind4, ResMap, UCSF Chimera (or ChimeraX if used).

We have added references for those software packages.

(19) METHODS>Image processing and 3D reconstruction> 2nd last paragraph: The complex samples are labelled sometimes with scFv16 and sometimes without. Please label them properly to avoid confusion.

We have revised this part accordingly.

(20) Extended Data Table 1>Refinement: There is no such a thing called “Model resolution”. A model is a model with static coordinates of each atom. Please remove the “Model resolution & FSC threshold” in the Refinement part. Instead, the authors can supply the statistics of Map-Model correlation.

We have deleted the rows of "Model resolution" and "FSC threshold". This information is redundant of map resolution. We also changed the row of "Map resolution (Å) " to "Map resolution (Å) (FSC=0.143)" to indicate how we calculated the resolution.

When all the remarks above are updated, I suggest Nature Communication to accept this

manuscript and publish this great work from Zhuang et al.

REVIEWERS' COMMENTS

Reviewer #1 (Remarks to the Author):

The revision to the manuscript "Molecular recognition of formylpeptides and diverse agonists by the formylpeptide receptors FPR1 and FPR2" by Zhuang et al has addressed all of my concerns. The resulting manuscript is well written and comprehensive. The new FRP1 and FRP2 structures are certainly of interest to a broad audience. Great job by Zhuang et al, and I look forward to this work being published.

Reviewer #2 (Remarks to the Author):

I thank all the authors for the input and correction and incorporate my suggestions into the updated manuscript. The revised version has satisfied all the points raised up in my previous feedback. I only would like to point out one little thing in the Figures 3C/5B/6B about the Ligplot presentation. Would it be possible to orient the ligands in the same way and direction as in Figure 3B/5A/6A?

For Figure 3C:

Ile5 should be pointing upwards and f-Met1 downwards.

(Left panel:) Rotating 180 degree would be perfect.

(Right panel:) it needs flipping vertically.

For Figure 5B:

Flip horizontally.

For Figure 6B:

Flip vertically.

Thank you.

Suggestions for Ligplot presentation

Figure 3B/C

Figure 5A/B

Figure 6A/B

Manuscript ID: NCOMMS-21-34674A

Title: **Molecular recognition of formylpeptides and diverse agonists by the formylpeptide receptors FPR1 and FPR2**

We thank all reviewers for their constructive comments. Please see our detailed responses to the comments below. The reviewers' comments are in **blue** font and our responses are in **black** font.

Reviewer #1 (Remarks to the Author):

The revision to the manuscript "Molecular recognition of formylpeptides and diverse agonists by the formylpeptide receptors FPR1 and FPR2" by Zhuang et al has addressed all of my concerns. The resulting manuscript is well written and comprehensive. The new FRP1 and FRP2 structures are certainly of interest to a broad audience. Great job by Zhuang et al, and I look forward to this work being published.

We thank the reviewer for the positive comments.

Reviewer #2 (Remarks to the Author):

I thank all the authors for the input and correction and incorporate my suggestions into the updated manuscript. The revised version has satisfied all the points raised up in my previous feedback. I only would like to point out one little thing in the Figures 3C/5B/6B about the Ligplot presentation. Would it be possible to orient the ligands in the same way and direction as in Figure 3B/5A/6A?

For Figure 3C:

Ile5 should be pointing upwards and f-Met1 downwards.

(Left panel:) Rotating 180 degree would be perfect.

(Right panel:) it needs flipping vertically.

For Figure 5B:

Flip horizontally.

For Figure 6B:

Flip vertically.

We thank the reviewer for the positive comments. We have modified the figures as suggested by the reviewer.